# Density Spectral Array Enables Precise Sedation Control for Supermicrosurgical Lymphaticovenous Anastomosis: A Retrospective Observational Cohort Study

**DOI:** 10.3390/bioengineering10040494

**Published:** 2023-04-21

**Authors:** En-Bo Wu, Yu-Hsuan Lin, Johnson Chia-Shen Yang, Chiung-Wen Lai, Jo-Chi Chin, Shao-Chun Wu

**Affiliations:** 1Department of Anesthesiology, Kaohsiung Chang Gung Memorial Hospital, College of Medicine, Chang Gung University, Kaohsiung 833, Taiwan; 2Division of Plastic and Reconstructive Surgery, Department of Surgery, Kaohsiung Chang Gung Memorial Hospital, College of Medicine, Chang Gung University, Kaohsiung 833, Taiwan; 3Department of Anesthesiology Jen-Ai Hospital, Taichung 412, Taiwan; 4Department of Anesthesiology, Park One International Hospital, Kaohsiung 813, Taiwan

**Keywords:** bispectral index, electroencephalographic density spectral array, late-elderly, lymphedema, non-intubated anesthesia, supermicrosurgical lymphaticovenous anastomosis

## Abstract

Supermicrosurgical lymphaticovenous anastomosis (LVA) is a minimally invasive surgical technique that creates bypasses between lymphatic vessels and veins, thereby improving lymphatic drainage and reducing lymphedema. This retrospective single-center study included 137 patients who underwent non-intubated LVA in southern Taiwan. A total of 119 patients were enrolled and assigned to two study groups: the geriatric (age ≥ 75 years, *n* = 23) and non-geriatric groups (age < 75 years, *n* = 96). The primary outcome was to investigate and compare the arousal and maintenance of the propofol effect-site concentration (Ce) using an electroencephalographic density spectral array (EEG DSA) in both groups. The results showed that the geriatric group required less propofol (4.05 [3.73–4.77] mg/kg/h vs. 5.01 [4.34–5.92] mg/kg/h, *p* = 0.001) and alfentanil (4.67 [2.53–5.82] μg/kg/h vs. 6.68 [3.85–8.77] μg/kg/h, *p* = 0.047). The median arousal Ce of propofol among the geriatric group (0.6 [0.5–0.7] μg/mL) was significantly lower than that in patients aged ≤ 54 years (1.3 [1.2–1.4] μg/mL, *p* < 0.001), 55–64 years (0.9 [0.8–1.0] μg/mL, *p* < 0.001), and <75 years (0.9 [0.8–1.2] μg/mL, *p* < 0.001). In summary, the combined use of EEG DSA provides the objective and depth of adequate sedation for extensive non-intubated anesthesia in late-elderly patients who undergo LVA without perioperative complications.

## 1. Introduction

The World Health Organization predicts that within the next 30 years, the proportion of the world’s population aged > 60 years will almost double from 12% to 22%. This indicates that the pace of population aging is accelerating faster than at any time in the past. According to estimates, the current global population of individuals aged 60 years or older has exceeded that of children aged 5 years or younger [1]. The term “elderly” is commonly defined as individuals who are 65 years of age or older based on chronological age. [2]. Individuals aged between 65 and 74 years are commonly referred to as “early-elderly adults,” while those aged ≥ 75 years are referred to as “late-elderly adults” [2]. Additionally, individuals aged ≥ 85 years are often classified as “oldest-old” or “extremely elderly adults” [3,4]. With age, the incidence of cancer-related lower limb lymphedema (LLL) is likely to increase [5,6,7,8,9,10] and can become highly severe with increased age-related comorbidities [11].

Surgical interventions for LLL include physiological restoration procedures, such as vascularized lymph node flap transfer (VLNT) and supermicrosurgical lymphaticovenous anastomosis (LVA). The conventional concept is that VLNT is indicated for high-grade lymphedema, while LVA is suitable for low-grade lymphedema. LVA is a minimally invasive surgical technique that creates bypasses between lymphatic vessels and veins, thereby improving lymphatic drainage and reducing lymphedema. LVA improves lymphedema by redirecting stagnant lymphatic fluid from the affected limb’s lymphatic vessels into the recipient’s veins [12,13,14,15,16,17]. In recent years, LVA has been shown to improve not only mild but also moderate-to-severe lymphedema [12,18,19].

LVA usually requires a prolonged operative time (>7 h in our institution) [20]. However, several trials have shown that a prolonged operative time is a crucial risk factor for postoperative delirium (POD) [21,22] and cognitive dysfunction (POCD) [23] among elderly patients. POD and POCD have been associated with a significantly prolonged stay, cognitive impairment, functional decline, and increased mortality at 6–12 months [23,24]. Recent studies have demonstrated that bispectral index (BIS) monitoring during anesthesia is associated with superior outcomes compared to non-BIS monitored anesthesia, such as a shorter time to extubation, faster emergence from anesthesia, and quicker recovery of orientation [25]. The bispectral monitor is a frontal electroencephalography-derived processed index (BIS™, Medtronic, Minneapolis, MN, USA). It directly measures the hypnotic effects induced by anesthetics in the brain. Electromyography (EMG) of BIS monitoring reflects the intensity of pain induced by surgical stress or involuntary movement in the absence of paralysis during surgery. Therefore, incorporating BIS monitoring during general anesthesia helps to maintain a highly precise and stable hypnotic status [26]. However, as the BIS value is not only the result of the integration of electroencephalography but also other processed variables [27], some interference in the BIS values has been reported, including the presence of EMG values [28] and increased delta power induced by noxious stimuli [29,30]. In total intravenous general anesthesia (TIVA), hypnotics, and opioids are not directly measured; however, the plasma concentration is calculated using a compartmental pharmacokinetic model built into an infusion pump. Extending the compartmental pharmacokinetic model used in pumps to predict plasma concentrations of hypnotics or opioids with an effect compartment enables the creation of a model of the effect-site concentration (Ce) [31,32]. With this approach, both agents can be administered to a target Ce through a technique known as target-controlled infusion (TCI). The pharmacokinetic model takes into account the patient’s age, weight, height, and sex [33], but its reliability may be limited in certain patient populations, such as the extremely elderly [34]. Because concentrations are calculated rather than measured, electroencephalography-based devices are recommended to monitor the depth of anesthesia in patients undergoing TIVA to avoid accidental awareness [35].

Given the increasing number of late-elderly adult patients with LLL, we examined the use of BIS-guided non-intubated anesthesia among these patients. Therefore, we conducted a single-center discovery study to investigate the arousal and maintenance Ce of propofol using an electroencephalographic density spectral array (EEG DSA) of BIS in late-elderly patients of the Taiwanese population who underwent LVA.

## 2. Materials and Methods

This retrospective observational cohort study was approved by the Institutional Review Board (IRB) of Kaohsiung Chang Gung Memorial Hospital (IRB number 202001420B0). This study used the Strengthening the Reporting of Observational Studies in Epidemiology (STROBE) statements, and the article was in accordance with applicable guidelines [36].

### 2.1. Patients

Data from 137 patients with LLL who underwent LVA at our institution in southern Taiwan between January 2018 and July 2020 were screened. In all cases, EEG DSA-guided intraoperative drug titration was used for the TCI of propofol and alfentanil to achieve precise non-intubated anesthesia. Medical records were retrieved from our database; patients with bilateral LLL, previous LVA, liposuction, the Charles procedure, loss of follow-up, or incomplete data were excluded. Eighteen patients were excluded: seven had bilateral LLL or previous surgery for LLL, five were lost to follow-up, and six had incomplete data. Finally, 119 patients were enrolled and assigned to two study groups: the geriatric group was defined as patients aged > 75 years (*n* = 23) and the non-geriatric group as patients aged < 75 years (*n* = 96) (Figure 1).

### 2.2. Evaluation of Lymphedema

Lymphedema was confirmed through lymphoscintigraphy prior to the surgery. The severity of lymphedema was classified using the International Society of Lymphology (ISL) staging system, with mild cases categorized as stages 0–1 and moderate-to-severe cases categorized as stages 2–3 [37]. Based on the ISL staging system for the surgical intervention of lymphosclerosis, it is generally considered that stage 0 and stage 1 (s0 and s1) are suitable for LVA, stage 2 (s2) is not recommended for LVA, and stage 3 (s3) can only be considered for VLNT. However, our previous study found that LVA is equally effective for moderate-to-severe lymphedema as it is for mild lymphedema. [12]. Table 1 shows the demographic data, including sex, age, etiology of LLL (classified as gynecologic or non-gynecologic cancer), ISL stage, body mass index (BMI), presence of diabetes mellitus or hypertension, affected lower limb side, adjuvant chemotherapy and radiotherapy, duration of lymphedema, cellulitis episodes, and volume gained in the affected limb, which were evaluated in this study. Among them, cases of gynecological cancer included cervical, endometrial, and ovarian cancers. Non-gynecological cancers included breast cancer, sarcoma, and lung cancer.

### 2.3. Intraoperative and Postoperative Management

During the intraoperative period, the following parameters were assessed under the surgical microscope: number of incisions per patient, number and size of lymphatic vessels found per patient, number of LVAs performed per patient (end-to-end or end-to-side configuration), NECST classification based on the appearance of the lymphatic vessels (normal, ectasis, contraction, or sclerosis) [38], total number and median diameter of recipient veins, and number of recipient veins per patient (Table 2). Ectasis is characterized by the flattening of lymphatic endothelial cells due to increased endolymphatic pressure. The contraction type is characterized by the transformation of lymphatic smooth muscle cells into synthetic cells that promote the growth of collagen fibers. Ultimately, in the sclerosing type, the lumen of the lymphatic vessels is narrowed or completely blocked by the accumulation of fibrous components [39].

All LVA procedures were conducted by a skilled surgeon, using a high-power surgical microscope (Pentero 900, Carl Zeiss AG, Oberkochen, Germany) and 11-0 nylon sutures (Ethilon, Ethicon, NJ, USA). The operative technique has been described in our previous study [12]. For postoperative care, all patients were instructed to wear custom-made compression stockings starting 1 week after the LVA surgery. They were to be worn at least during daytime activities. The duration of post-LVA follow-up and the amount and percentage of volume reduction in both groups were also assessed (Table 2). In addition, the volume gained in the lymphedematous limb was measured using magnetic resonance (MR) volumetry and calculated as the difference between the volume of the affected limb and the volume of the contralateral normal limb.

### 2.4. MR Volumetry for Lower Extremities

The MR examination for measuring lymphedema volume was conducted with the patient in the supine position. A single Siemens MAGNETOM Skyra 3T MRI scanner with two 18-channel body matrix coils (Siemens Healthcare GmbH, Erlangen, Germany) was used for MR volumetry. For LLL, anatomical T1-weighted images of bilateral lower legs were obtained using a coronal three-dimensional sampling perfection with application-optimized contrasts using different flip angle evolutions (SPACE, Siemens) with the following parameters: repeat time/echo time, 500–622/11 ms; field of view, 40 cm; matrix size, 320 × 320; voxel size, 1.3 × 1.3 × 3.0 mm^3^; 60 contiguous slices without inter-slice gap. The volume of the lower extremities was calculated using the commercially available AZE Virtual Place software (AZE Ltd., Tokyo, Japan). The detailed protocol for this procedure has been described in our previous publication [12].

### 2.5. Anesthetic Management

All patients were induced with a BIS value guided by slow propofol titration (the effect-site targeting mode of the Schnider model) and maintained at a BIS value of 45–65 as an adequate depth of anesthesia. For BIS-guided non-intubated anesthesia with TCI (Fresenius Orchestra Infusion Workstation; Fresenius Kabi, Taipei, Taiwan), the operative time, respiratory rate, oxygen saturation, urine output, BIS value, propofol and alfentanil consumption, maintenance Ce of propofol and alfentanil, and arousal Ce of propofol and alfentanil were recorded for both groups (Table 3). The alpha power (8–12 Hz) was preserved in DSA (BIS™ monitoring system software version 3.50, Medtronic, Minneapolis, MN, USA) throughout the entire period of anesthesia (Figure 2), allowing for the precise control of hypnotic drugs (propofol) and rapid recovery without complications.

Opioids can restore alpha power (reversing alpha dropout) and prevent delta band arousal (reducing power in the delta band [0.5–4 Hz]) caused by surgical nociceptive input [40]. Therefore, alfentanil (the effect-site targeting mode of the Scott model) was administered according to the EMG value and EEG DSA of the BIS monitoring in the range of 15–40 ng/mL. Supplemental humidified oxygen (6 L/min) was administered through a simple mask, and end-tidal carbon dioxide was continuously monitored. Positioning the head and neck in an ideal position is the key to non-intubation techniques with spontaneous ventilation. The upper airway patency was set at a 15° head-down tilt, neck roll pillow, nasopharyngeal airway device, and headrest that kept the chin plane horizontally aligned (Figure 3). The pulse oximeter was set as the beat-to-beat mode instead of the default mode (average value in previous 15 s measurements). The respiratory rate and respiratory pattern (waveform) were continuously monitored using the impedance of the electrocardiogram. The combination of spirometry and impedance of the electrocardiogram could help promptly identify central respiratory depression, peripheral airway obstruction, or mixed hypoventilation. Inadequate respiration can be reversed by reducing propofol or alfentanil concentrations with BIS guidance. Intraoperative fluid management was performed using goal-directed fluid therapy: patients received a starting dose of lactated Ringer’s solution at 4 mL/kg/h via intravenous infusion. Continuous noninvasive hemodynamic monitoring using a finger cuff (ClearSight system, Edwards Lifesciences, Irvine, CA, USA) was used to adjust fluid administration for all patients.

### 2.6. Primary and Secondary Outcomes

The primary outcome was to investigate and compare the arousal and maintenance of propofol Ce using EEG DSA in elderly and late-elderly patients who underwent LVA. In addition, we classified the elderly group into three subgroups (<54-years-old, 55–64-years-old, and 65–74-years-old) and compared them with the late-elderly group (>75-years-old). Five additional intraoperative secondary outcomes were assessed and compared between both groups, including the operation time, intraoperative respiratory rate, pulse oximetry, BIS values, and urine output. Other secondary outcomes included postoperative volume reduction, nausea and vomiting (PONV), delirium, cognitive dysfunction, respiratory complications, and cardiovascular events.

### 2.7. Statistical Analysis

The normality of continuous numeric data was assessed using the Kolmogorov–Smirnov normality test. Student’s *t*-test was used to compare normally distributed variables, presented as the mean ± standard deviation. The Mann–Whitney U test was used to compare non-parametric variables, which were presented as medians (interquartile range [IQR]). Categorical variables, including sex, etiology, lymphosclerosis classification, comorbidities, and previous cancer treatment, were analyzed using the Chi-square or Fisher’s exact tests. Post-hoc analysis was performed using one-way analysis of variance with Bonferroni correction. IBM SPSS^®^ Statistics version 22.0 was used for data analysis. Statistical significance was defined as a *p*-value < 0.05.

## 3. Results

### 3.1. Demographic Data

A total of 23 patients (21 females and 2 males) were enrolled in the geriatric group, while 96 patients (93 females and 3 males) were enrolled in the non-geriatric group. Patients in the geriatric group were significantly older (79.4 ± 2.7 years vs. 59.2 ± 9.4 years, *p* < 0.001), had received less hypertensive treatment (15 out of 23 vs. 37 out of 96, *p* = 0.021), had received less chemotherapy (4 out of 23 vs. 42 out of 96, *p* = 0.02), and had a higher median number of cellulitis episodes (2 [1–3] vs. 1 [0–2], *p* = 0.027) compared to those in the non-geriatric group. There were no significant intergroup differences in terms of sex (*p* = 0.247), etiology of lower limb lymphedema (gynecologic or non-gynecologic cancer, *p* = 0.173), ISL stage (*p* = 0.328), BMI (*p* = 0.392), diabetes mellitus (*p* = 0.360), affected side of the lower limb (left or right, *p* = 0.947), radiotherapy (*p* = 0.724), median duration of lymphedema (*p* = 0.06), or median volume gained in the lymphedematous limb (*p* = 0.286) (Table 1).

### 3.2. Intraoperative Findings

During LVA, 155 and 753 lymphatic vessels were identified in Groups I and II, respectively (*p* < 0.001). Between the two groups, no significant differences were found in median incisions (*p* = 0.380), median lymphatic vessels found per patient (*p* = 0.123), number (*p* = 0.887) and size (*p* = 0.837) of recipient veins per patient, and median number of LVAs performed per patient (*p* = 0.314). The number of LVAs performed was higher than the number of identified lymphatic vessels because it is sometimes possible to create antegrade and retrograde anastomoses using a single lymphatic vessel. In addition, the geriatric group had a larger median overall lymphatic vessel diameter (0.8 [0.5–1.0] mm vs. 0.6 [0.4–0.8] mm, *p* < 0.001) and showed a significant difference in NECST classification compared to the non-geriatric group (*p* < 0.001) (Table 2).

### 3.3. Surgical Outcomes

The average follow-up period after LVA was 10.6 months for both groups (*p* = 0.988). There was no significant difference in the volume reduction after LVA in milliliters (513.0 [178.0–1080.0] ml vs. 684 [101.5–1505.5] ml, *p* = 0.614) or in the percentage reduction between the two groups (19.7 [8.0–60.2]% vs. 34.1 [6.9–62.1]%, *p* = 0.681) (Table 2). In either group, no surgery-related complications, such as wound dehiscence, hematoma, or delayed wound healing, were observed.

### 3.4. BIS-Guided Non-Intubated Anesthesia

The median operative time (463.0 [389.0–504.0] min vs. 451.5 [385.0–511.0] min, *p* = 0.593), respiratory rate (11 [9–12.5] vs. 11 [10–13], *p* = 0.987), oxygen saturation (99 [98, 99, 100]% vs. 99 [98, 99, 100], *p* = 0.953), and propofol Ce (2.0 [1.5–2.2] μg/mL vs. 2.0 [1.8–2.5] μg/mL, *p* = 0.24) were similar between the two groups (Table 3). Urine output was significantly low in the geriatric group (1.59 [0.84–2.70] ml/h/kg vs. 2.58 [1.78–3.79] ml/h/kg, *p* = 0.012), but still better than the usual normal range set at 0.5–1.0 mL/h/kg. The median BIS value was similar between the two groups (55 [49–58] vs. 55 [50–60], *p* = 0.773), indicating that an adequate and similar anesthetic depth was achieved. The geriatric group required less propofol (4.05 [3.73–4.77] mg/kg/h vs. 5.01 [4.34–5.92] mg/kg/h, *p* = 0.001) and alfentanil (4.67 [2.53 –5.82] μg/kg/h vs. 6.68 [3.85–8.77] μg/kg/h, *p* = 0.047) than the non-geriatric group. No differences were found in the Ce for propofol (2.0 [1.5–2.2] μg/mL) vs. 2.0 [18–2.5] μg/mL), *p* = 0.240) and alfentanil (18 [15–30] ng/mL vs. 25 [25–30] ng/mL, *p* = 0.114) between the two groups. When comparing the median arousal Ce of propofol among patients in the geriatric group (0.6 [0.5–0.7] μg/mL), it was significantly high in patients aged ≤ 54 years (1.3 [1.2–1.4] μg/mL, *p* < 0.001), 55–64 years (0.9 [0.8–1.0] μg/mL, *p* < 0.001), and <75 years (0.9 [0.8–1.2] μg/mL, *p* < 0.001). However, in patients aged 65–74 years, no differences were found in the arousal Ce of propofol (0.7 [0.6–0.8] μg/mL, *p* = 0.079) (Figure 4). No PONV, POD, POCD, respiratory complications, or cardiovascular events were observed in either group.

## 4. Discussion

In this retrospective observational cohort study, we compared the arousal Ce of propofol and consumption of propofol in elderly and late-elderly Taiwanese patients who underwent non-intubated LVA. To our knowledge, this is the first study to investigate the use of BIS-guided non-intubated anesthesia with TCI in a time-consuming procedure (median time, 463.0 min), particularly among late-elderly patients.

Except for some minimally invasive procedures under general anesthesia [41], according to the literature review, the non-intubated technique with spontaneous ventilation is primarily applied to thoracic surgery with/without a supraglottic airway device and is accompanied by regional nerve blockade [42,43]. Non-intubated techniques in video-assisted thoracic surgery show fast postoperative recovery, low complication rates, short postoperative fasting times and hospital stays [44], and reduced stress and inflammatory responses [45]. However, the literature suggests that the practice of non-intubated anesthesia for thoracic surgery still considers patient selection, anesthetic preparation, and other potential perioperative complications [45].

We introduced BIS monitoring with EEG DSA into our anesthesia technique without additional regional blocks and titrated the dosage of hypnotics and analgesics according to the interpretation of EEG DSA and BIS. This way, the elderly patient maintained spontaneous breathing with a permissive hypercapnia state during the operation.

The median arousal Ce of propofol was significantly lower in the geriatric group than in the non-geriatric group. Propofol consumption per unit of body weight per hour in the geriatric group was also significantly reduced. This trend using BIS monitoring shows real-world evidence of decreased propofol use among older adult patients. The analysis revealed that the arousal Ce of propofol decreased significantly with age, as shown in Figure 4. Further subgroup analysis was conducted to explore the potential differences in propofol’s effect on arousal across different age groups. Among the subgroups, the geriatric group had a significantly lower arousal Ce of propofol than patients aged ≤ 54 years and 55–64-years-old. As individuals age, the concentration of propofol required for anesthesia decreases. This is due to the fact that the aging brain is more sensitive to the effects of propofol [33]. Moreover, aging and anesthesia both cause a decrease in hepatic blood flow, leading to a reduced ability of the elderly liver to rapidly clear propofol from the system [46]. As a result, the combination of these two effects can increase the sensitivity of elderly individuals to propofol by up to 50%, as reported in previous studies [47]. In our study, we found that the lower Ce of propofol and alfentanil during emergence in the geriatric group may be associated with these aforementioned findings, and there may be a synergistic effect when propofol and alfentanil are used in combination [48].

POD is a well-known risk factor associated with advancing age. Meta-analyses have shown that the use of BIS-guided general anesthesia in older adult patients can reduce POD by up to 29–49% [26,38,49] and postoperative cognitive dysfunction by up to 16–31% [26,49]. A multicenter research trial indicated that a lack of intraoperative alpha spindles in electroencephalography was associated with delirium in the post-anesthesia care unit [50]. Klimesch and Purdon et al. showed that alpha frequency [51] and intraoperative alpha power [52] linearly decrease with age. In our study, alfentanil was used for analgesia and could prevent or recover an alpha dropout [53]. In addition, under the guidance of the BIS, we avoided burst suppression or shortened its duration as much as possible [54]. Neither POD nor POCD was found in our research groups, although both groups included elderly patients.

Under the guidance of a BIS value of 45–65, both Groups I and II achieved an adequate hypnotic status. Regarding noxious stimulation, EEG DSA has three representations [55]: (1) the increase in beta wave power (12–25 Hz) known as beta arousal is typically observed when a low dose of hypnotics is combined with insufficient analgesia and is often accompanied by somatic movement; (2) a sudden and episodic decrease in frontal alpha wave power, referred to as alpha dropout, is indicative of a nociceptive stimulus and is commonly observed during body surface surgery; (3) delta arousal (delta wave power increased) with or without alpha dropout, which is often observed as a response to noxious stimulation of the body cavity. Therefore, if the depth of anesthesia is determined solely based on the BIS value, it is considered a possible misjudgment. The BIS value is affected by certain situations. For example, vibrations caused by forced-air warming blankets, increasing EMG values (over 30–40 Hz) [56] due to involuntary movement without a neuromuscular blocking agent [57], or delta arousal caused by noxious stimuli will cause an increase in the BIS value [55]. Therefore, we suggest that the interpretation of the EEG DSA findings is crucial during BIS monitoring.

Precision anesthesia is a balanced concept that includes hypnosis, analgesia, muscle relaxation, and hemodynamic status, which are monitored separately and integrated into the anesthesia plan [58]. In our study, precision anesthesia relied on the anesthesiologist to precisely judge the current state of the brain based on the results of EEG DSA and decide to use hypnotics or analgesics instead of the conventional method of judging the administration of anesthetics according to secondary outcomes, such as blood pressure and/or heart rate. For example, EEG DSA changes, such as alpha dropout or delta arousal, are often best treated with increased analgesics rather than hypnotics [55]. In our study, a significant alpha dropout implied noxious stimulation, especially during intraoperative indocyanine green injection and skin incision. To the best of our knowledge, proper analgesic management during surgery is critical. Nevertheless, the potential influence of neurophysiological and immune responses to noxious stimuli during surgery on long-term patient outcomes has not been fully elucidated and warrants further investigation in large-scale clinical trials.

A significant limitation of our study is that the patient groups had a relatively low BMI (27.5 ± 5.6 vs. 26.5 ± 5.0 kg/m^2^), which is common among Asians. Theoretically, EEG DSA-guided anesthesia involves the direct interpretation of the processed EEG and is not affected by race or BMI. However, there is insufficient evidence to support this claim. Further studies are needed in non-Asian populations or those with higher BMIs to determine the universal applicability of our EEG DSA-guided anesthesia research results. An additional important limitation to relying solely on BIS monitoring of EMG and EEG DSA is the absence of objective pain monitoring, such as nociceptive level (NOL^®^, Medasense Biometrics Ltd., Ramat Gan, Israel) monitoring [59], which is not yet accessible in Taiwan. The dosage of alfentanil depends mainly on the experience of the anesthesiologist. Finally, LVA is a minimally invasive surgery with relatively small wounds and shallow dissection (usually <1 cm in depth), which is ideal for BIS-guided non-intubated anesthesia. More evidence is needed to determine whether more invasive and painful procedures, such as abdominal, chest, or orthopedic surgery, are suitable for BIS-guided non-intubated anesthesia.

## 5. Conclusions

The combined use of EEG DSA provides an objective and adequate depth of sedation for extensive non-intubated anesthesia in late-elderly patients who underwent LVA without perioperative complications.

## Figures and Tables

**Figure 1 bioengineering-10-00494-f001:**
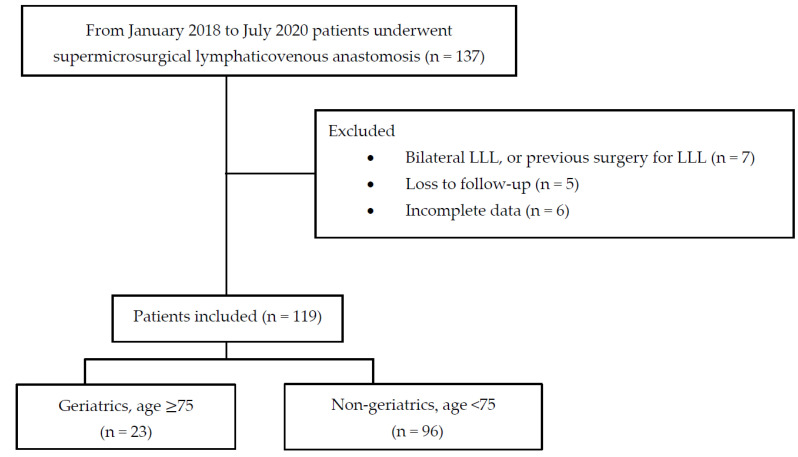
Flow diagram of case selection. LLL, lower limb lymphedema.

**Figure 2 bioengineering-10-00494-f002:**
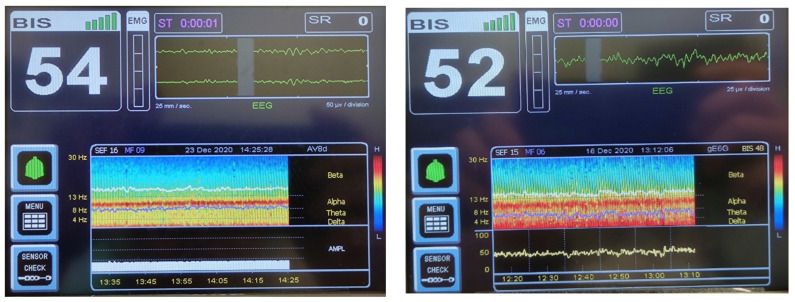
(**a**) (**Left**) and (**b**) (**right**). An example of intraoperative BIS monitoring with EEG DSA used in a 78-year-old female in the geriatric group (Figure 2a) and a 47-year-old female in the non-geriatric group (Figure 2b). The depth of anesthesia is similar, with adequate BIS values between 45 and 65. Both DSAs show that the alpha power (8–12 Hz) remained constant throughout the entire period of anesthesia, allowing for precise control of the hypnotic drug (propofol) and rapid recovery without complications. BIS, bispectral index; EEG DSA, electroencephalographic density spectral array.

**Figure 3 bioengineering-10-00494-f003:**
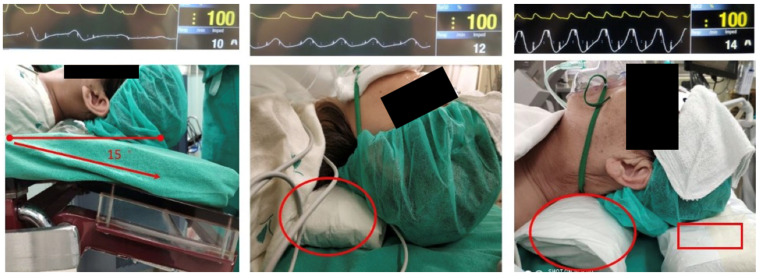
Upper airway position of patients for non-intubating anesthesia. Left: 15° head-down tilt only. Middle: 15° head-down tilt with a neck roll pillow (indicated by the red circle). Right: 15° head-down tilt with a neck roll pillow and keeping the chin plane horizontally aligned by a headrest (indicated by the red square) to be able to optimize respiratory impedance.

**Figure 4 bioengineering-10-00494-f004:**
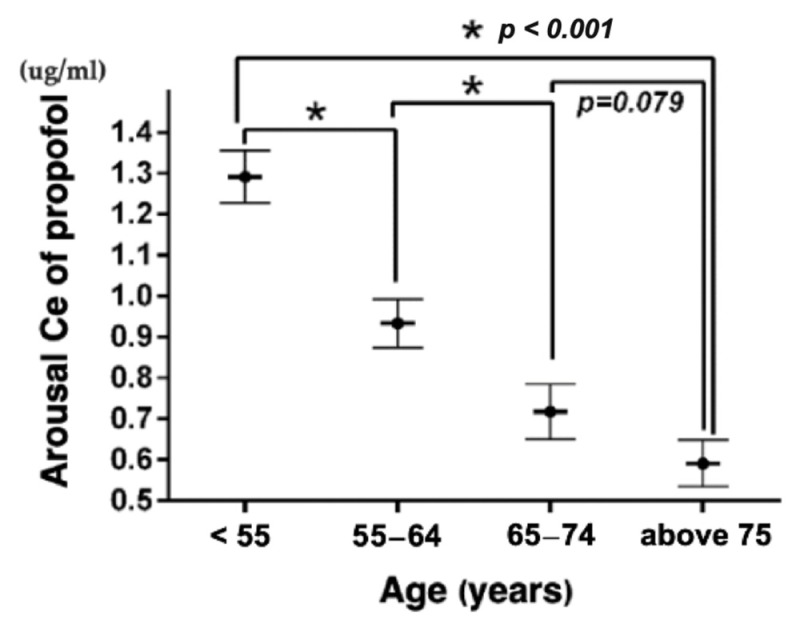
Comparison of the arousal effect-site concentration of propofol among different age groups using subgroup analysis. Ce, effect-site concentration.

**Table 1 bioengineering-10-00494-t001:** Demographic and clinical characteristics of the patients.

Variables (Unit)	Geriatrics (Age ≥ 75, *n* = 23)	Non-Geriatrics (Age < 75, *n* = 96)	*p*-Value
Female/male	21/2	93/3	0.247
Age (year)	79.4 ± 2.7	59.2 ± 9.4	<0.001
Etiology of LLL, *n* (%)			0.173
Gynecologic cancer *	18 (78.3%)	68 (70.8%)	
Non-gynecologic cancer **	5 (21.7%)	28 (29.2%)	
ISL stage (0–1/2–3)	2/21	4/92	0.328
BMI (kg/m^2^)	27.5 ± 5.6	26.5 ± 5.0	0.392
Diabetes mellitus, yes/no	2/21	19/77	0.360
Hypertension, yes/no	15/8	37/59	0.021
Affected lower limb (left/right)	13/10	55/38	0.947
Chemotherapy, yes/no	4/19	42/54	0.020
Radiotherapy, yes/no	11/12	42/54	0.724
Duration of lymphedema (year)	9.6 (3.4–13.7)	4.2 (1.3–10.2)	0.06
Cellulitis episode (per year)	2 (1–3)	1 (0–2)	0.027
Volume gained in the lymphedematous limb ^@^ (mL)	2325.0 (1709.0–3921.7)	2123.5 (1548.5–2923.5)	0.286

Student’s *t*-test was used to compare normally distributed variables, presented as the mean ± standard deviation. Non-normally distributed data are expressed as the median [inter-quartile range (IQR)]. LLL, lower limb lymphedema; ISL, International Society of Lymphology; BMI, body mass index. * Gynecologic cancer cases included cervical, endometrial, and ovarian cancers. ** Non-gynecologic cancer included patients with breast cancer, sarcoma, and lung cancer. ^@^ Equals preoperative lymphedema limb volume minus contralateral normal limb volume.

**Table 2 bioengineering-10-00494-t002:** Intraoperative and postoperative surgical presentation.

Variables (Unit)	Geriatrics (Age ≥ 75, *n* = 23)	Non-Geriatrics (Age < 75, *n* = 96)	*p*-Value
Intraoperative			
Incisions (*n*)	4 (3–4)	4 (3–5)	0.380
Lymphatic vessels found (*n*)	5 (4–6)	5 (4.5–7)	0.123
Size of lymphatic vessels (mm)	0.80 (0.5–1.0)	0.60 (0.45–0.80)	<0.001
Total number of recipient’s veins (*n*)	85	433	
Recipient veins per patient (*n*)	4 (3–5.5)	4 (3–5)	0.887
Size of recipient veins per patient (mm)	0.80 (0.7–1.0)	0.80 (0.6–1.0)	0.837
Number of LVA performed per patient (*n*)	7 (5.5–8.5)	8 (5–10)	0.314
NECST classification, *n* (%)			<0.001
Normal	62.40%	384.51%	
Ectasis	53.34%	113.15%	
Constriction	31.20%	226.30%	
Sclerosis	9.6%	30.4%	
Postoperative			
Mean follow-up, post-LVA (month)	10.6 ± 5.8)	10.6 ± 6.0	0.988
Post-LVA volume reduction * (mL)	513 (178–1080)	684 (101.5–1505.5)	0.614
Post-LVA percentage of volume Reduction ** (%)	19.7 (8.0–60.2)	34.1 (6.9–62.1)	0.681

Student’s *t*-test was used to compare normally distributed variables, presented as the mean ± standard deviation. Non-normally distributed data are expressed as the median [inter-quartile range (IQR)]. LVA, supermicrosurgical lymphaticovenous anastomosis. * Mean post-LVA volume reduction (mL) = V_preoperative − V_postoperative, where V_preoperative is the preoperative lymphedematous limb volume and V_postoperative is the postoperative lymphedematous limb volume. ** Mean post-LVA volume reduction (%) = ((V_pre − V_post)/V_gain) × 100, where V_pre is the preoperative lymphedematous limb volume in mL, V_post is the postoperative lymphedematous limb volume in mL, and V_gain is the volume gained in the lymphedema limb in mL.

**Table 3 bioengineering-10-00494-t003:** BIS-guided non-intubation total intravenous anesthesia with target-controlled infusion.

Variables (Unit)	Geriatrics (Age ≥ 75, *n* = 23)	Non-Geriatrics (Age < 75, *n* = 96)	*p*-Value
Operation time (min)	463.0 (389.0–504.0)	451.5 (385.0–511.0)	0.593
Respiratory rate (/min)	11 (9–12.5)	11 (10–13)	0.987
SpO_2_ (%)	99 (98–100)	99 (98–100)	0.953
Urine output (ml/kg/h)	1.59 (0.84–2.70)	2.58 (1.78–3.79)	0.012
Bispectral index (BIS value)	55 (49–58)	55 (50–60)	0.773
Propofol			
Total consumption (mg/kg/h)	4.05 (3.73–4.77)	5.01 (4.34–5.92)	0.001
Maintenance Ce by Schnider model (μg/mL)	2.0 (1.5–2.2)	2.0 (1.8–2.5)	0.240
Arousal Ce (μg/mL)	≥75 years, 0.6 (0.5–0.7)	0.9 (0.8–1.2)	<0.001
		≤54 years, 1.3 (1.2–1.4)	<0.001
		55–64 years, 0.9 (0.8–1.0)	<0.001
		65–74 years, 0.7 (0.6–0.8)	0.079
Alfentanil			
Total consumption (μg/kg/h)	4.67 (2.53–5.82)	6.68 (3.85–8.77)	0.047
Maintenance Ce by Scott model (ng/mL)	18 (15–30)	25 (20–30)	0.114
Arousal Ce (ng/mL)	15 (10–30)	25 (20–30)	0.026

Non-normally distributed data are expressed as the median [inter-quartile range (IQR)]. BIS, bispectral index; Ce, effect-site concentration.

## Data Availability

The data presented in this study are available from the corresponding author upon reasonable request.

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
