# Peer review of "Density Spectral Array Enables Precise Sedation Control for Supermicrosurgical Lymphaticovenous Anastomosis: A Retrospective Observational Cohort Study"

_bioengineering, 2023, doi:10.3390/bioengineering10040494_

Round 1

Reviewer 1 Report

I appreciate the authors presenting this clinical application useful research article emphasizing the role of use of EEG DSA in for extensive non-intubated anesthesia in late-elderly patients who underwent LVA. The rich are well support the hypothesis. My  only comments is : 

Since  groups I and 2 have the same depth of anaesthesia (BIS =55 median) (Table 3) Why Group 1 clearly has a lower arousal Ce for prpopfol and alfentanil. Please make a brief discussion.

Reviewer 2 Report

Title: “Density Spectral Array Enables Precise Sedation Control for Supermicrosurgical Lymphaticovenous Anastomosis: A Retrospective Observational Cohort Study

Abstract: Supermicrosurgical lymphaticovenous anastomosis (LVA) is a minimally invasive bypass surgery for lymphatic vessels that improves lymphedema. This retrospective single-center study included 137 patients who underwent non-intubated LVA in southern Taiwan. A total of 119 patients were enrolled and assigned to two study groups: the geriatric (age75 years, n = 23) and non-geriatric groups (age <75 years, n = 96). The primary outcome was to investigate and compare the arousal and maintenance of the propofol effect-site concentration (Ce) using an electroencephalographic density spectral array (EEG DSA) in both groups. The results showed that the geriatric group required less propofol (4.05 [3.734.77] mg/kg/h vs. 5.01 [4.345.92] mg/kg/h, p = 0.001) and alfentanil (4.67 [2.535.82] μg/kg/h vs. 6.68 [3.858.77] μg/kg/h, p = 0.047). The median arousal Ce of propofol among the geriatrics group (0.6 [0.50.7] μg/ml) was significantly lower than patients aged ≤54 years (1.3 [1.21.4] μg/ml, p < 0.001), 5564 years (0.9 [0.81.0] μg/ml, p < 0.001), and <75 years (0.9 [0.81.2] μg/ml, p < 0.001). In summary, the combined use of EEG DSA provides objective and depth of adequate sedation for extensive non-intubated anesthesia in late-elderly patients who underwent LVA without perioperative complications.

General comment: Although the topic of this work seems to be interesting some issues should be reworked to enhance the quality and the impact of the main text. In the following some detailed comment:

Detailed comments:

1) Figure 1. Flow diagram of case selection. *Geriatrics is defined as >75 years of age. LVA, lymphati-covenous anastomosis; LLL, lower limb lymphedema.

*) Figure 1 should be reworked and improved.

2) Figure 3 lines: “Figure 3. Upper airway position of patients for non-intubating anesthesia. Left: 15ï‚° head-down tilt only. Middle: 15ï‚° head-down tilt with a shoulder roll (indicated by the red circle). Right: 15ï‚° head-down tilt with a shoulder roll and keep the chin plane horizontally aligned by a headrest (indicated by the red square) to be able to optimize respiratory impedance.

*) This figure should be improved and the privacy of patients should be respected.

*)Figure 4 should improved together with its caption.

*) tables 1,2,3 should be made more clear to the interested readers. Please rework.

Lines” One major limitation of this study is that our patient groups had a relatively low BMI 364
(27.5 ± 5.6 vs. 26.5 ± 5.0 kg/m2), which is common among Asians. Whether our findings are 365
generalizable to patient groups with a high BMI will require further research in non-Asian 366
populations. Another important limitation to the sole use of BIS monitoring of EMG and 367
EEG DSA is the lack of objective pain monitoring, such as nociceptive level (NOL®, 368
Medasense Biometrics Ltd., Ramat Gan, Israel) monitoring [56], which is currently not 369
available in Taiwan. The dosage of alfentanil depends mainly on the experience of the 370
anesthesiologist. Finally, LVA is a minimally invasive surgery with relatively small 371
wounds and shallow dissection (usually <1 cm in depth), which is ideal for BIS-guided 372
non-intubated anesthesia. More evidence is needed to determine whether more invasive 373
and painful procedures, such as abdominal, chest, or orthopedic surgery, are suitable for 374
BIS-guided non-intubated anesthesia.

*) The authors should better describe the value of this work for “non-Asian” populations. Please rework and improve.

The Quality of English is sufficient
